# Prevalence and Risk Factors of Central Obesity among Adults with Normal BMI in Shaanxi, China: A Cross-Sectional Study

**DOI:** 10.3390/ijerph182111439

**Published:** 2021-10-30

**Authors:** Wen-Yu Feng, Xiang-Dong Li, Juan Li, Yuan Shen, Qiang Li

**Affiliations:** 1Department of Epidemiology and Biostatistics, School of Public Health, Xi’an Jiaotong University, Xi’an 710061, China; fwy999@stu.xjtu.edu.cn (W.-Y.F.); lijuan15180219@163.com (J.L.); yulander.s@xjtu.edu.cn (Y.S.); 2School of Public Health, Xi’an Jiaotong University, Xi’an 710061, China; lxd941115@stu.xjtu.edu.cn

**Keywords:** central obesity, body mass index, prevalence, risk factor, metabolic disease

## Abstract

(1) Background: The study aims to estimate the prevalence of normal weight with central obesity (NWCO) and to examine the relationship between NWCO and cardiovascular disease risk factors in adults of the province of Shaanxi. (2) Methods: A population-based cross-sectional survey was conducted among residents who were aged 18–80 years and had been living in Zhenba County, Shaanxi Province, for over six months in 2018. Descriptive data analysis and prevalence/frequency were conducted. Logistic regression analyses were used to detect the corresponding factors associated with central obesity. (3) Results: A total of 2312 participants (936 men and 1376 women) were analyzed. The prevalence of NWCO was 58.3%. NWCO was significantly associated with hypertension and dyslipidemia. Compared with normal weight non-central obesity (NWNO), the adjusted odds ratios (ORs) for hypertension were 1.47 (95% CI 1.10–1.98) in men and 1.55 (1.14–2.10) in women, and the corresponding odds ratios for dyslipidemia were 2.71 (1.77–4.13) in men and 1.84 (1.29–2.61) in women. Female sex, age over 58 years, and lower education level were also significantly predictors of abdominal obesity. (4) Conclusions: Body mass index alone as a measure of obesity is not sufficient for assessing health risks. Central obesity index should be used together for clinical assessment.

## 1. Introduction

Obesity is a medical condition characterised by an abnormal fat accumulation, which may have adverse effects on health, thereby reducing life expectancy [1]. Excess body weight is the sixth most important risk factor contributing to the overall burden of disease worldwide [2]. In 2016, more than 1.9 billion adults aged 18 years and older were overweight. Of these, over 650 million adults were obese [3]. Overweight and obesity have become a major public health problem in China [4,5]. A recent study showed that abdominal obesity is associated with insulin resistance and higher risks of metabolic syndrome and cardiovascular diseases in Asian females, whereas general obesity is not [6]. Central obesity has been recognised as an independent risk factor for cardio-metabolic diseases and a better predictor of cardiovascular risk than overall obesity [7,8]. Defined by normal body mass index (BMI) and higher waist-to-height ratio, normal weight with central obesity (NWCO) has been recognised as a risk factor for cardio-metabolic diseases [9]. The most common form of obesity is usually assessed by BMI [10], which has the main limitation of not differentiating body fat from lean mass and central from peripheral fat. Therefore, athletes with enhanced body muscle mass may be misclassified as obese when using only BMI to diagnose obesity, whereas people with low lean mass but high body fat content may still have a normal BMI [11]. By contrast, proxy measures of anthropometric indicators such as waist circumference (WC), waist-to-hip ratio (WHR), and waist-to-height ratio (WHtR) are used to assess central fat distribution [12]. WHtR is a better predictor than BMI and WC for diabetes, dyslipidemia, hypertension, and cardiovascular disease (CVD) [13]. WHtR may be a simpler and better predictor of early health risks [14,15,16,17].

This study aims to estimate the prevalence of normal weight central obesity classified by using BMI and WHtR and to examine the relationship between normal weight with central obesity and CVD risk factors in adults in Shaanxi, China.

## 2. Materials and Methods

Subjects: A population-based cross-sectional survey was conducted among residents who were 18–80 years old and had been living in Zhenba County, Shaanxi Province, for over six months in 2018. The multistage stratified cluster sampling method was used to enroll the study’s participants. Six districts in Zhenba County, Shaanxi Province, were randomly selected, and then six communities were randomly selected from each district. All residents in the communities were invited to join this survey. Patients who were psychotic, debilitated, pregnant, or handicapped in any form to the point that obtaining anthropometric measurement would be difficult were excluded from the study. A consecutive sample of 3781 subjects aged over 18 years was recruited. Subjects with incomplete data due to height, weight, waist circumference, hypertension, dyslipidemia, and diabetes were excluded. A total of 2312 normal BMI participants were chosen for the study.

Data Collection: Data were obtained by face-to-face interviews composed of three parts: questionnaire investigation (included socio-demographic characteristics and health-related information), body measurements (such as height, weight, WC, and blood pressure), and laboratory measurements (such as serum cholesterol and triglyceride). Demographic variables including sex, age, level of education, cigarette smoking, coffee consumption, tea intake, and alcohol drinking were assessed by self-reporting on the use of any tobacco product or alcohol drinking in the past years. All investigators were trained by research assistants to collect data through the same questionnaire instruction.

Measurements: Height and weight were measured in increments of 0.1 cm and 0.1 kg, respectively, by trained staff. BMI was calculated as the weight (kg) divided by the squared height (m^2^). WC was measured to the nearest 0.1 cm at the umbilical level in a standing position. WHtR was calculated as WC divided by height. Omron automatic blood pressure machine was used to determine the subjects’ blood pressure on the right arm after at least 5 min of seated rest. Blood pressure was measured two times with intervals of at least one minute. The average value was used for data analysis. Fasting venous blood samples were drawn from the study subjects in order to measure serum levels of high-density lipoprotein cholesterol (HDL-C), low-density lipoprotein cholesterol (LDL-C), triglycerides, and blood glucose. The samples were stored in a cooler at 4 °C for transportation to the Department of Laboratory Medicine, the First People’s Hospital of Zhenba County, and measured within 24 h of being drawn.

Definition of Related Indicators: BMI was categorised in accordance with the WHO’s criteria [3] as underweight (<18.5 kg/m^2^), normal (18.5–24.9 kg/m^2^), overweight (25.0–29.9 kg/m^2^), and obese (>30.0 kg/m^2^). Central obesity was defined according to the WHO criteria [18,19]: WHtR < 0.5 (non-central obesity) and ≥0.5 (central obesity). Normal-weight central obesity was defined as central obesity in participants with normal weight (by BMI). The subjects were categorized into the following two groups: normal weight non central obesity (NWNO); and normal weight central obesity (NWCO). Underweight and overweight subjects (BMI < 18.5 and BMI ≥ 25.0 kg/m^2^) were excluded from the study. Self-reported hypertensions and/or abnormal blood pressure (systolic ≥ 140 mmHg or diastolic ≥ 90 mmHg) were regarded as hypertension [20]. The patient who had a history of hypertension and was currently taking medication for hypertension was also defined as having hypertension. Self-reported diabetes mellitus (DM) and/or a fasting serum glucose level ≥ 7.0 mmol/L were/was regarded as DM [21]. The patient who had a history of diabetes and was currently taking medication for diabetes was also defined as having diabetes. According to the criteria of the “Chinese Guidelines on Prevention and Treatment of Dyslipidemia in Adults,” hyperlipidemia diagnosed by a physician and/or abnormal blood lipids (TC ≥ 6.2 mmol/L or TG ≥ 2.3 mmol/L or HDL-C < 1.0 mmol/L) were defined as hyperlipidemia [22].

Statistical Methods: Data were expressed as mean values ± standard deviations (SD) for continuous variables. Counts (frequencies = *n*) and proportions (%) were reported for categorical variables. Chi-squared test and Fisher’s exact probability method were used to compare categorical variables. The significant variables were included in the binary logistic regression and adjusted for confounding factors. Analysis was carried out at a 95% confidence level. Data were analyzed by the complex sampling function of SPSS 23.0 (IBM, New York, NY, United States), and a *p* value of < 0.05 was considered statistically significant.

## 3. Results

### 3.1. Univariate Analysis of Associated Factors of Normal Weight Central Obesity

In the study, we analyzed a total of 2312 participants (936 men and 1376 women). The mean age of the participants was 58.2 ± 13.1 years. Age groups were grouped according to a mean age of 58 years. The majority of the participants were elder people and females. Of the 2312 participants classified as normal weight using BMI, the prevalence of central obesity was 58.3% by WHtR. The characteristics of the two groups classified by WHtR (NWNO and NWCO) are shown in Table 1. Among people with normal body mass index, the average age of NWCO group and NWNO group was 60.5 ± 12.1 years and 55.1 ± 13.9 years, respectively. The older the age, the higher the prevalence of central obesity. The prevalence in NWCO was 50.3% in men and 63.7% in women, respectively. The prevalence of NWCO was 68.3%, 74.2%, and 66.2% in hypertension, dyslipidemia, and diabetes, respectively. Central obesity was significantly associated with hypertension and dyslipidemia but not significant for DM among participants with normal weight, measured by BMI. Education, cigarette smoking, and coffee consumption were significantly associated with central obesity using WHtR. There were no significant correlations between central obesity and tea intake or alcohol drinking (Table 1).

### 3.2. Logistic Regression Analysis of Associated Factors of Normal Weight Central Obesity

However, after adjusting for confounding factors (diabetes, alcohol drinking, tea intake, and coffee consumption), only female, age over 58 years, cigarette smoking, education, hypertension, and dyslipidemia were significantly associated with central obesity among the study participants. Female participants were 1.66 times more likely to be central obese than their male counterparts (Table 2).

Logistic regression analysis was conducted in order to calculate the ORs (95% CIs) for hypertension, dyslipidemia, and other factors in each gender (Table 3 for men and Table 4 for women). When compared with NWNO, the adjusted ORs for hypertension (adjusted OR 1.47, 95% CI 1.10–1.98 in men; 1.55, 1.14–2.10 in women) and dyslipidemia (2.71, 1.77–4.13 in men; 1.84, 1.29–2.61 in women) had statistically significant associations in NWCO. In comparison with no cigarette smoking, daily cigarette smoking was associated with lower risk of NWCO. Adjusted OR for NWCO was 0.52 in males, 95% CI 0.40–0.68, *p* < 0.001. Among females, there were no statistically significant associations with NWCO for either cigarette smoking or no cigarette smoking.

## 4. Discussion

In our study, the prevalence of NWCO defined using a combination of BMI and WHtR was 58.3%. As people become older, the risk of central obesity increases gradually. The prevalence of NWCO was higher in women than in men (63.7% vs. 50.3%). Normal weight with central obesity was associated with CVD risks such as hypertension and dyslipidemia in the present study, regardless of sex.

The higher prevalence of normal-weight central obesity found in this study suggests the need to include anthropometric indices in the measurement of excessive body weight other than BMI alone, as BMI alone might result in misclassifications and underestimation of at-risk individuals. This being the case, it is important to conduct screening for NWCO by using a combination of BMI and WHtR [23]. Such individuals would normally need to be offered the appropriate health education and prompt intervention to manage and/or prevent the development of cardio-metabolic complications. At the same time, it also provides a new perspective for prevention of metabolic diseases: pay attention to your WHtR even when you have a normal BMI.

Only female sex, age over 58 years, cigarette smoking, education, hyperlipidemia, and hypertension were associated with central obesity. The association of central obesity with age and sex has been documented by several studies [24]. The higher likelihood of being obese among females can be linked to women’s lower engagement in physical activity as well as the physiological changes that occur during their reproductive years [25,26]. Moreover, the distribution of fat in the body also changes after middle age: body fat is more likely to accumulate in the abdomen [27]. This could also be a plausible reason for the higher prevalence of obesity among older participants. An inverse association was found between cigarette smoking and central obesity. In comparison with no cigarette smoking, daily cigarette smoking had an inverse association with the risk of NWCO. This is consistent with previous studies [28,29]. In addition, the cumulative effect of cigarette smoking on health is detrimental, regardless of its effect on weight loss. The majority of the study participants had low level education. There was significant correlation between central obesity and educational background. Central obesity has been shown to be associated with low education and socioeconomic status, which are characterised by poor health behaviour [30,31].

Hypertension and hyperlipidemia had a positive association with central obesity, which is consistent with previous studies [32]. People with normal BMI but increased WC have a higher risk of developing metabolic diseases. Moreover, a study has also shown that measures of central obesity are more strongly associated with total and cardiovascular disease death than BMI [16]. Central obesity was found to be strongly related to certain metabolic risk factors among non-obese subjects [33]. In our study, although the prevalence of diabetes was higher in obese patients with normal BMI, no significant correlation between central obesity and hyperglycemia was detected. Our study demonstrates that normal-weight central obesity is associated with hypertension and hyperlipidemia. A previous study noted that central obesity among normal-weight individuals has been shown to be associated with a greater cardiovascular risk and mortality than is found among normal weight individuals without central obesity [16]. Thus, blood glucose, blood lipid, and blood pressure should be detected early among adults with normal BMI in order to reduce the prevalence of hypertension, diabetes, and hyperlipidemia. Measuring waist circumference should be recommended as a simple and efficient tool for screening central obesity and related metabolic risks even in non-obese individuals. This highlights the need to include measures of central obesity in all clinical assessments in this study setting. Measures aimed at reducing obesity should be prioritised in order to curb the growing prevalence of chronic diseases, which impacts greatly the already over-burdened healthcare system.

The study still has some limitations. First, the cross-sectional nature of this study and the use of self-reported lifestyle behaviour inevitably produced recall bias. Second, the participants were recruited from Zhenba County in Shaanxi Province; thus, the conclusion cannot represent the situation in other regions of China. Thus, further studies, including prospective studies, will be required in order to establish causality. Finally, the sample size of this study was limited, and more confounding variables could not be adjusted.

## 5. Conclusions

The prevalence of central obesity is high in adults of normal weight. Sex, age, education, and cigarette smoking were closely related to central obesity among the study’s participants. The study suggests that using BMI alone as a measure of obesity is not enough for assessing health risks. In clinical practice, WHtR for measuring central obesity should be considered.

## Figures and Tables

**Table 1 ijerph-18-11439-t001:** Comparison of the prevalence of central obesity in participants under different groups of associated factors (Chi-square test are applied).

	NWCO	NWNO	*χ^2^*	*p*
Age				
<58	525 (50.9)	507 (49.1)	41.853	<0.001
≥58	822 (64.2)	458 (35.8)		
Sex				
Male	471 (50.3)	465 (49.7)	40.780	<0.001
Female	876 (63.7)	500 (36.3)		
Hypertension				
Yes	424 (68.3)	197 (31.7)	35.025	<0.001
No	923 (54.6)	768 (45.4)		
Dyslipidemia				
Yes	256 (74.2)	89 (25.8)	42.379	<0.001
No	1091 (55.5)	876 (44.5)		
Diabetes				
Yes	96 (66.2)	49 (33.8)	4.016	0.045
No	1251 (57.7)	916 (42.3)		
Tea intake				
Yes	725 (57.8)	529 (42.2)	0.224	0.636
No	622 (58.8)	436 (41.2)		
Coffee consumption				
Yes	26 (44.8)	32 (55.2)	4.415	0.036
No	1321 (58.6)	933 (41.4)		
Alcohol drinking				
Yes	266 (56.7)	203 (43.3)	0.577	0.447
No	1081 (58.7)	762 (41.3)		
Cigarette smoking				
None	1092 (61.7)	677 (38.3)	50.871	<0.001
Few times	43 (64.2)	24 (35.8)		
Sometimes	10 (71.4)	4 (28.6)		
Daily	202 (43.7)	260 (56.3)		
Education				
Primary school and below	584 (66.2)	298 (33.8)	47.496	<0.001
Junior middle school	461 (55.4)	371 (4.6)		
Senior middle school	201 (48.9)	210 (51.1)		
College	84 (58.7)	59 (41.3)		
High than college	17 (38.6)	27 (61.4)		

Data are expressed as *n* (%). NWNO: normal weight non-central obesity; NWCO: normal weight central obesity.

**Table 2 ijerph-18-11439-t002:** Logistic regression showing predictors of normal weight central obesity.

Variables	β	Wald χ^2^	*p*	OR (95%CL)
Age ≥ 58	0.52	26.39	<0.001	1.69 (1.38, 2.06)
Sex = Female	0.46	15.43	<0.001	1.59 (1.26, 2.00)
Hypertension	0.43	15.71	<0.001	1.53 (1.24, 1.89)
Dyslipidemia	0.79	32.48	<0.001	2.19 (1.68, 2.88)
Cigarette smoking		27.65	<0.001	
None				1.00
Few times	0.22	0.61	0.433	1.24 (0.72, 2.13)
Sometimes	0.44	0.52	0.473	1.56 (0.46, 5.24)
Daily	−0.67	23.10	<0.001	0.51 (0.39, 0.67)
Education		10.37	0.035	
Primary school and below				1.00
Junior middle school	−0.18	2.91	0.088	0.83 (0.68, 1.03)
Senior middle school	−0.34	6.31	0.012	0.71 (0.55, 0.93)
College	0.03	0.02	0.884	1.03 (0.70, 1.52)
High than college	−0.68	4.09	0.043	0.51(0.26,0.98)

Adjusted for diabetes, tea intake, alcohol drinking, and coffee consumption. OR, odds ratio; 95% CI, 95% confidence interval.

**Table 3 ijerph-18-11439-t003:** Logistic regression showing predictors of normal weight central obesity in males.

Variables	β	Wald χ^2^	*p*	OR (95%CL)
Age ≥ 58	0.58	13.13	<0.001	1.78 (1.30, 2.44)
Hypertension	0.39	6.56	0.010	1.47 (1.10, 1.98)
Dyslipidemia	1.00	21.25	<0.001	2.71 (1.77, 4.13)
Cigarette smoking		21.70	<0.001	
None				1.00
Few times	0.19	0.33	0.564	1.21 (0.64, 2.29)
Sometimes	0.24	0.15	0.704	1.28 (0.36, 4.49)
Daily	−0.64	18.28	<0.001	0.53 (0.39, 0.71)
Education		9.84	0.043	
Primary school and below				1.00
Junior middle school	−0.02	0.01	0.915	0.98 (0.70, 1.37)
Senior middle school	−0.15	0.58	0.448	0.86 (0.58, 1.27)
College	0.76	6.42	0.011	2.14 (1.19, 3.86)
High than college	−0.24	0.25	0.619	0.79 (0.30, 2.04)

Adjusted for diabetes, tea intake, alcohol drinking, and coffee consumption. OR, odds ratio; 95% CI, 95% confidence interval.

**Table 4 ijerph-18-11439-t004:** Logistic regression showing predictors of normal weight central obesity in females.

Variables	β	Wald χ^2^	*p*	OR (95%CL)
Age ≥ 58	0.49	12.47	<0.001	1.62 (1.24, 2.13)
Hypertension	0.44	7.87	0.005	1.55 (1.14, 2.10)
Dyslipidemia	0.61	11.37	0.001	1.84 (1.29, 2.61)
Education		11.64	0.020	
Primary school and below				1.00
Junior middle school	−0.28	3.83	0.050	0.76 (0.58, 1.00)
Senior middle school	−0.45	5.86	0.015	0.64 (0.44, 0.92)
College and above	−0.59	4.59	0.032	0.56 (0.33, 0.95)
High than college	−1.09	5.13	0.023	0.34 (0.13, 0.86)
Cigarette smoking		4.08	0.253	

Adjusted for diabetes, tea intake, alcohol drinking and coffee consumption. OR, odds ratio; 95% CI, 95% confidence interval.

## Data Availability

The datasets used and/or analyzed during the current study are available from the corresponding author upon reasonable request.

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
