# Peer review of "Prevalence and Risk Factors of Central Obesity among Adults with Normal BMI in Shaanxi, China: A Cross-Sectional Study"

_ijerph, 2021, doi:10.3390/ijerph182111439_

Round 1
Reviewer 1 Report
The manuscript clearly demonstrates benefits of monitoring central obesity rather than depending on BMI alone to predict health risks particularly those associated with CVD. I recommend the paper is accepted for publication with very minor polishing up.
Lines 112 and 113, it is stated that the prevalence of central obesity among men and women is 50.3% and 63.7% respectively, this is not reflected in Table 1 unlike the other parameters, rather the data given are the percentages of men and women out of the sample selected for the study i.e. 1347 while the prevalence of central obesity among men and women is calculated as (471/936)% and (876/1376)% respectively.
Author Response
Response to Reviewer 1 Comments
On behalf of my co-authors, we appreciate reviewers very much for your positive and constructive comments and suggestions on our manuscript. We have read the comments carefully and have made corrections which we hope meet with approval. Revised portion are marked up using the “Track Changes” function in the paper.
Lines 112 and 113, it is stated that the prevalence of central obesity among men and women is 50.3% and 63.7% respectively, this is not reflected in Table 1 unlike the other parameters, rather the data given are the percentages of men and women out of the sample selected for the study i.e. 1347 while the prevalence of central obesity among men and women is calculated as (471/936)% and (876/1376)% respectively.
Response 1:
Line 114: modify the percentages in males and females in Table 1 to the prevalence of NWCO in different sex groups, the other groups were modified in turn.
We appreciate for Reviewers’ warm work earnestly, and hope that the correction will meet with approval. Once again, thank you very much for your comments and suggestions.
Kind regards,
Yours sincerely,
Li Qiang
Reviewer 2 Report
The aim of the manuscript written by Feng and collaborators, was to perform a population-based cross-sectional survey among residents living in the Shaanxi Province, to estimate the prevalence of Normal weight with central obesity (NWCO) and to examine the relationship between NWCO and cardiovascular disease risk factors. The analysis performed led the authors conclude that BMI alone as a measure of obesity is not enough to assess health risks and central obesity index should be also used for clinical assessment. Indeed, the prevalence of NWCO was significantly associated with hypertension and dyslipidemia, and furthermore female sex (over 58 years) and lower education level were significantly predictors of abdominal obesity.
In my opinion, the paper is interesting, well written and very easy to read. Although it presents some limitations, as reported by the authors in the discussion, I think it provides a comprehensive analysis demonstrating why there should not be used the only BMI as measure of obesity. Having say that, I only have some minor point to suggest.
Minor points:
- Line 7 and 8: please add the initials for the Xiang-dong Li and Qiang Li, respectively.
- Line 18: add a space after “1.55”.
- Line 34: add a space after “is not”.
- Line 39: please combine the 2 sentences: The most common form of obesity is usually assessed by BMI [10] which has the main limitation to not differentiate….
- Line 58: no need to repeat “town”. Only “district” is fine. The same for “villages” with “communities”.
- There were exclusion criteria other than underweight and overweight subjects? If yes please indicate in the Methods section.
- Line 72: please replaced “automated” with “automatic”.
- Line 81: add the word “criteria” or “standards” after “WHO”.
- Line 111: is obvious the meaning of NWNO but you have to define since is the first time you are using.
- Tables: in all tables’ captions, should be better explain the type of analysis performed.
In Table 1 is the p value representing the significance between the only NWCO with the considered variable?
Also, always indicate the standard deviation in the same way ±…
Also, in Tables 2 and 3 there is no value in the row “cigarette smoking none”, and in Tables 2, 3 and 4 there is no value in the row “education primary school and below”.
- Line 163: The “P” of Pay should be lowercase.
- Line 170: The “B” of Body should be lowercase.
